# Generating controlled gust perturbations using vortex rings

**Dipendra Gupta**[1,2], **Sanjay P. Sane**[3]*, **Jaywant H. Arakeri**[2]*

**1** Sibley School of Mechanical and Aerospace Engineering, Cornell University, Ithaca, New York, United States of America, **2** Department of Mechanical Engineering, Indian Institute of Science, Bangalore, India, **3** National Centre for Biological Sciences, Tata Institute of Fundamental Research, Bangalore, India

\* sane@ncbs.res.in (SPS); jaywant@iisc.ac.in (JHA)

## Abstract

To understand the locomotory mechanisms of flying and swimming animals, it is often necessary to develop assays that enable us to measure their responses to external gust perturbations. Typically, such measurements have been carried out using a variety of gusts which are difficult to control or characterize owing to their inherently turbulent nature. Here, we present a method of generating discrete gusts under controlled laboratory conditions in the form of a vortex rings which are well-characterized and highly controllable. We also provide the theoretical guidelines underlying the design of gust generators for specific applications. As a case study, we tested the efficacy of this method to study the flight response of freely-flying soldier flies *Hermetia illucens*. The vortex ring based method can be used to generate controlled gusts to study diverse phenomena ranging from a natural flight in insects to the artificial flight of insect-sized drones and micro-aerial vehicles.

## Introduction

Flying animals often move in erratic environments facing discrete gusts or continuous turbulence [1–3]. Such gusts are experienced when there is a sudden and sharp change in wind speed, as typically encountered in the wakes of large objects or at the edge of convective disturbances. Organisms may also encounter continuous turbulence when flying in open environments or at great heights which is usually described using statistical approaches. Despite these unpredictable conditions, flying animals including insects, birds or bats can successfully control their flight in face of strong gusts [4–10]. For several years, researchers from various fields have tried to quantify the locomotory abilities of diverse animals under challenging conditions. For instance, ecologists are interested in understanding how locomotory ability sets the range over which animals can disperse or migrate [11]. Neurobiologists are interested in the mechanistic details of how animals may sense, process and respond to perturbations to their trajectories [12]. More recently, the field of bioinspired robotics has posed similar questions about the flight ability of their flappers and drones or swimming robots [13].

A key requirement for such studies is the ability to generate a well-characterized gust perturbation which can be used to determine how stable the flying or swimming animals or a robot is in face of a sudden change in ambient conditions. To achieve this, researchers have used several methods to generate turbulence and gusts in context of natural fliers. These

**Data Availability Statement:** The data are present at this link: https://drive.ncbs.res.in/index.php/s/Bk7TF2EEjr46YJj.

**Funding:** Funding for this study was provided by grants from the Air Force Office of Scientific Research (AFOSR) # FA2386-11-1-4057 and #

FA9550-16-1-0155, and National Centre for Biological Sciences(Tata Institute of Fundamental Research) to SPS. We also acknowledge the support of the Ministry of Earth Sciences, Government of India, under project no. MESO-0034 and the Department of Atomic Energy, Government of India, under project no. 12-R&D-TFR-5.04-0800.

**Competing interests:** The authors have declared that no competing interests exist.

include grid-generated turbulence [4, 7, 14], von-Karman vortices [5, 15, 16], compressed air jet [10, 17, 18] etc. These studies have provided keen insights into the general ways in which animals respond to turbulent gusts. Other methods are focused on either flow characterization alone or on the passive response of airfoils or drones [19–21].

Here, we present a method that allows the delivery of precise and repeatable gusts, thereby enabling better control on the nature of perturbations encountered by animals in such experiments, and thus gain insight into the nature of their responses. The device presented here is capable of generating discrete, well-characterized, customized gusts using vortex rings. Vortex rings are typically generated by imparting an impulsive motion to a piston [22]. Vortex rings propagate with their own, self-induced velocity due to vorticity concentrated in their core region. A vortex ring is particularly effective in generating precise gusts, because of its sharp impulse and because vorticity is concentrated primarily in its core. This makes it possible to generate a high-speed perturbations while keeping the ring laminar, thereby eliminating any ambiguity in the flow characteristics. When turbulent gusts such as jets, grids, or bluff objects are placed upstream to perturb the flow to measure the response of the animals, their behaviour is a function of both mean flow and turbulent fluctuations making it difficult to interpret their response. In contrast to these methods, vortex rings are structured and relatively free from these effects. The flow physics of such a perturbation method is also well-understood, and hence the flow properties are highly controllable [23]. This method can be readily adapted to a variety of contexts ranging from aerial to aquatic locomotion. In this paper, we specifically use it to measure the responses of a freely flying soldier fly, *Hermetia illucens* as an example.

Vortex-generated gusts are different from freestream perturbations (e.g. von-Karman vortices) in which gusts superimpose over mean ambient flow, and which are well-suited for hovering insects. In contrast, the method presented here can be used to study the response of both hovering as well as freely-flying insects. Similar methods have been previously employed to study the response of hovering hawkmoths [24]. Here, we first provide a detailed description of the experimental setup to create such vortex rings and related gusts, and the spatio-temporal characterization of their flow properties, in addition to a detailed characterization of the flow properties in application to the animal locomotion which has been typically missing. As a case study, we next test this device on freely-flying soldier flies whose flight was perturbed using the gusts generated by this method. The wing span and the wing-tip speed of the insects dictated the size and velocity of the vortex ring in this study. We also outline the necessary theoretical details to estimate the flow properties of a vortex ring.

## Methods and materials

The most common method of generating a vortex ring is using a piston-cylinder arrangement (Fig 1A). The size of the rings depends on the exit diameter ($D_0$) of the cylindrical tube and the extent to which the piston has traversed (referred to as *stroke length*, *L*) which can be equal to, smaller or larger than the exit diameter of the tube. The moving piston drives a slug of fluid, causing vorticity to be generated in the boundary layer due to the no-slip condition. As the high-speed slug of fluid emerges from the nozzle, this boundary layer forms a cylindrical vortex sheet that rolls up into a spiral form, thus forming a vortex ring (Fig 1B–1D). In the formative stages, the vortex ring entrains the surrounding fluid as it propagates. This generates a vortex bubble ($D_{vb}$), the diameter of which is larger than the ring ($D_r$) (Fig 1E). Even after the formation process, entrainment of surrounding fluid can occur, followed by subsequent detrainment, the balance between which prevents a substantial change in its diameter.

The momentum (more precisely, *impulse*) of the ring is determined by the momentum imparted to the surrounding fluid in the tube by the piston. It depends on the type of motion

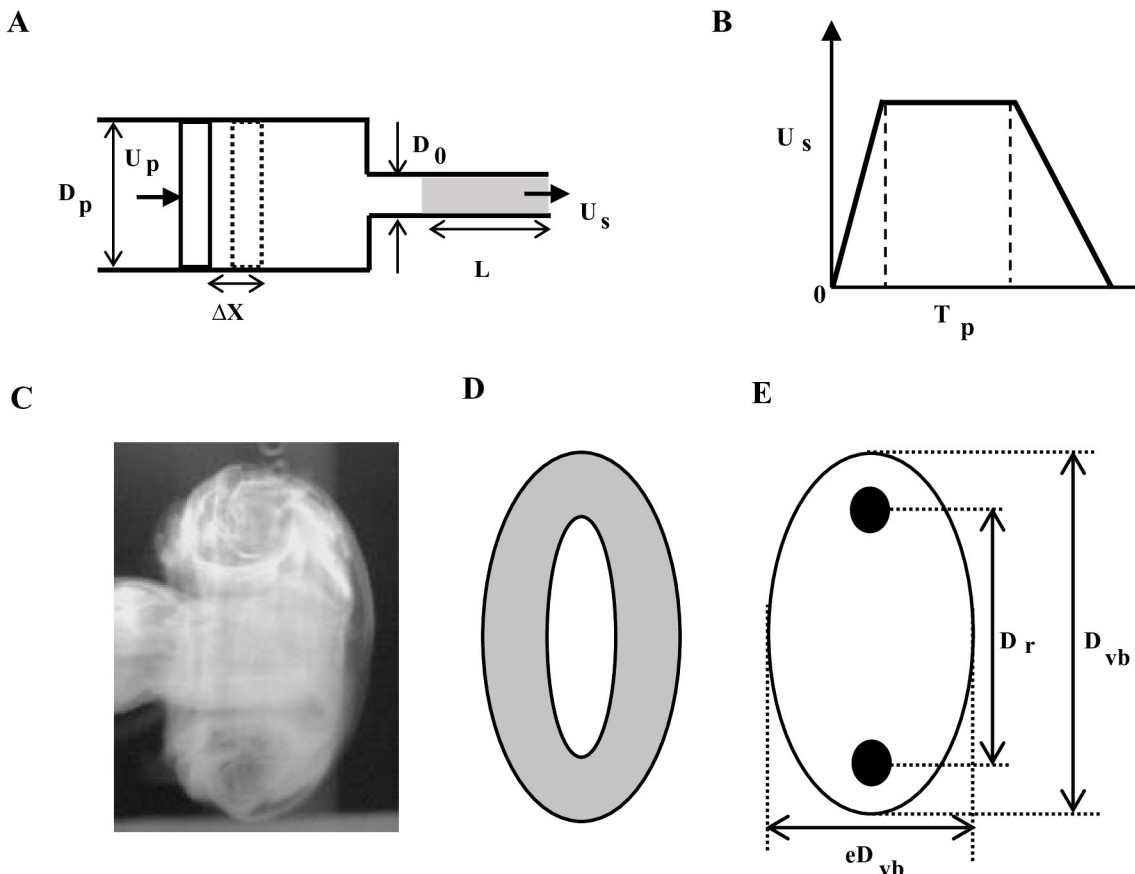

**Fig 1. Geometric details of vortex ring and ring generator.** (A) Piston-nozzle arrangement for generating vortex ring. Dashed line shows the outline of piston after it has moved by $\Delta X$. Grey shaded region is the length of fluid that emerges from the nozzle, called here as slug length ($L$). (B) Variation of piston velocity/slug velocity with time. $D$ and $U_p$ are diameter and velocity of piston, respectively, $D_0$ and $U_s$ are exit diameter and velocity at the exit of the nozzle, respectively. $T_p$ is the total time for which piston moves. (C) Side view, (D) isometric view and (E) line diagram of side view of the ring. Black circle denotes the core of the vortex ring, in which the vorticity is concentrated. $D_r$, $D_{vb}$ and $e$ denote diameter of the ring (distance between the cores), diameter of the vortex bubble including entrained air, and eccentricity of the ellipsoid, respectively.

imposed on the piston, fluid viscosity, piston travel time, its radius, and circulation. During vortex formation, the ring accelerates and then may rise to a constant velocity or slow down. The slowing down may be attributed to the entrainment of surrounding fluid, entrainment followed by detrainment, viscous diffusion of the core or vortex instabilities [25]. The turbulent vortex ring, as compared to the laminar one, is characterized by a rapid growth rate of its diameter and shedding of vorticity to the wake, which causes a rapid decrease in its propagation velocity. Thus, piston movement time (stroke time, $T_p$) and slug length ($L$) can be designed to generate the flow properties of the ring (Fig 1B). General working relations based on simple conservation laws to achieve this objective are provided in the Appendix (see S1 Appendix).

Our experimental set-up consisted of a 60 cm long, 30cm square cross-section dismountable clear Perspex chamber, a 40 cm long, 3.7 cm internal diameter ($D_o$) cylindrical PVC nozzle (2mm thick), a 12-inch 100W and 8Ω speaker, a digital-to-analog converter NI-DAQ, a high voltage high current direct coupled (DC) amplifier and a high-speed camera (Miro EX4, Vision Research, Ametek) (Fig 2). The large Perspex chamber served as a closed test section where we generated vortex ring. The larger dimensions of the test chamber compared to the

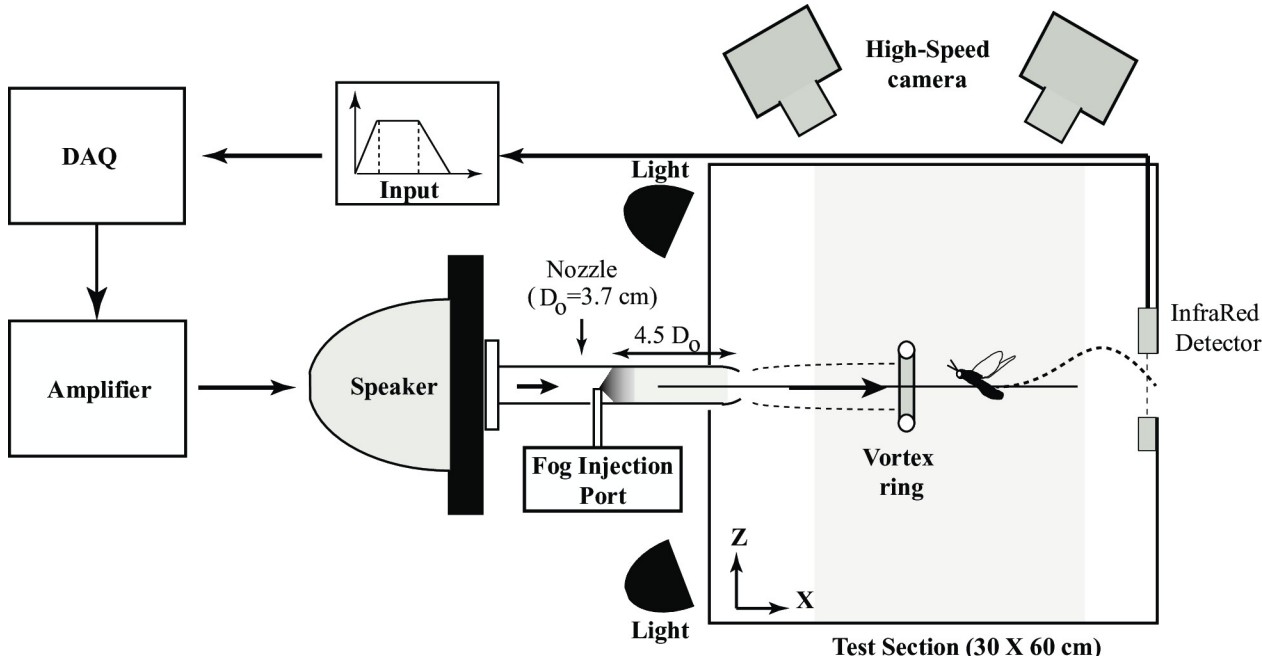

**Fig 2. Vortex ring generator system.** The input signal to the DAQ (top) is a trapezoidal wave with voltage amplitude V, rising time t1, constant duration t2-t1 and fall time t3-t2-t1. This signal is converted into analog form, amplified and fed to the speaker for vortex ring generation. Fog particles were injected into nozzle through fog injection port (FIP), and a high speed camera was placed horizontal to record the lateral view of the vortex ring as it propagates. In experiments with flies, we used two high-speed cameras to capture the 3D trajectories for the flies. All dimensions are in cm.

diameter of the vortex ring reduced any effects of ambient air currents on experimental observations, and hence aided in maintaining a still ambient fluid [26]. The experiments were carried out in a closed room with controlled humidity at an ambient temperature of 20˚C.

Instead of a piston-cylinder arrangement, we used a speaker to generate the vortex ring. The speaker was enclosed in a 40cm x 40cm x 5cm wooden chamber (driving section) on the diaphragm side, and each side of the chamber was glued with Fevicol[TM] (Pidilite, Mumbai, India) and pin hammered to ensure that it was airtight. A 5cm diameter hole was cut at the centre of the 40cm x 40cm face of the wooden chamber to facilitate attachment of the nozzle via a PVC flange. A rubber gasket placed between the flange and the wooden chamber eliminated any air leakage. The nozzle was sharp chamfered by an angle of 9˚ at the exit and smooth chamfered at entry. The speaker attached to the nozzle was then fitted to the test chamber through a 4cm hole cut on its longest side (Fig 2).

## Input signal

We first synthesized a trapezoidal signal (Fig 1B) using NI-LabVIEW, which consisted of three parts: acceleration (100 μs), constant velocity (30 ms), and deceleration (100 ms), such that it resulted in the desired velocity of the vortex ring. A large deceleration time eliminates the formation of stopping vortex, which results from the abrupt stopping of the piston, due to separation and rolling of the secondary boundary layer induced by the primary vortex ring on the outer surface of the tube [26]. When generating a gust, the formation of stopping vortex must be eliminated, because it alters the strength and size of the vortex ring [26], and also because it may induce a secondary response in insects which are hit first by the vortex ring and next by the stopping vortex. We converted the signal into analog form for physical output using NI–

c9263, amplified it using an in-house DC power amplifier, and fed it to the speaker, resulting in the formation of a vortex ring at the exit of the nozzle.

## Formation of vortex ring

The signal, when fed to the speaker, displaces the speaker diaphragm which imparts its momentum to the surrounding air, causing an equivalent volume of air being pushed out from the chamber into the nozzle. Unlike conventional piston-cylinder configuration for vortex ring generation [26], in our experimental set-up, we have used a long nozzle (11 times its diameter) so that the exit is far away from the speaker diaphragm. This eliminates the generation of a piston vortex and any disturbances similar to those generated using orifice [26].

## Characterization of vortex ring

The actual response of the insects depends on their radial position at the time they intercept the vortex ring (see ref. [27] for instantaneous streamlines). Although the vortex ring has an average self-induced translational velocity, the fluid associated with it has local axial and radial velocities, and these velocities depend on the radial location relative to the centre of the ring. Like translational velocity and diameter, the local velocities are well-documented and repeatable [23]. Here, we measured the gross flow properties of the vortex rings using two techniques: fog visualization and styrofoam bead method. Both methods were carried out separately, and flow properties were characterized using each method (see S1 Video).

## Fog visualization

We used fog particles to seed the flow (Antari Fogger, Taiwan) for visualization of the vortex ring. The average particle size was on the order of $1$–$2$ $\mu$m. The fog was first filled into a $500 ml$ wash bottle and injected through a fog injection port (FIP) on the nozzle (Fig 2). The port was made on the upper circumference of the nozzle, at 18 cm ($4.5 D_0$) away from its exit plane. The circumferential (lower, upper, or sidewise) position of the port did not affect the visualization. Its longitudinal position, however, determined whether any fog particles were present at the exit of the nozzle before ring formation (i.e. background fog at nozzle exit). Keeping FIP at this distance ensures there is no leakage of fog in the test chamber before the initiation of the diaphragm motion. A 5cm square window was hinged on the longer side opposite to nozzle. We closed the window during visualization to eliminate any effects of external air current on the ring propagation and its trajectory. After the recording, we opened the window to remove the residual fog inside the chamber before starting the next trial.

The vortex ring can be generated under varying Reynolds numbers (= $U_{avg}D_0/v$) (Fig 3). For example, we generated vortex rings at Reynolds numbers of $4.7{\times}10^3$ (Fig 3B) and $1.6{\times}10^4$ (Fig 3C). The vortex sheet emanates from the edge of the nozzle on triggering the speaker, rolls up to form a spiral and hence, a ring. This rolled up vortex propagates while drawing more fluid from the ambient surrounding until it attains a fully formed size. We did not observe any secondary and piston vortices (also [26]), which makes it well-suited for generating discrete gusts. Although the flow within the core of the ring produced here is laminar, it is also possible to produce turbulent rings by increasing their speed, diameter, or both [27, 28].

## Estimation of flow speed using styrofoam bead method

In addition to fog, we used a styrofoam bead to measure flow speeds. Because the density of styrofoam beads (6.52 kg/m³) is of the same order of magnitude as the density of air (for details, see [29]), the bead is expected to attain the same velocity as the gust that it intercepts.

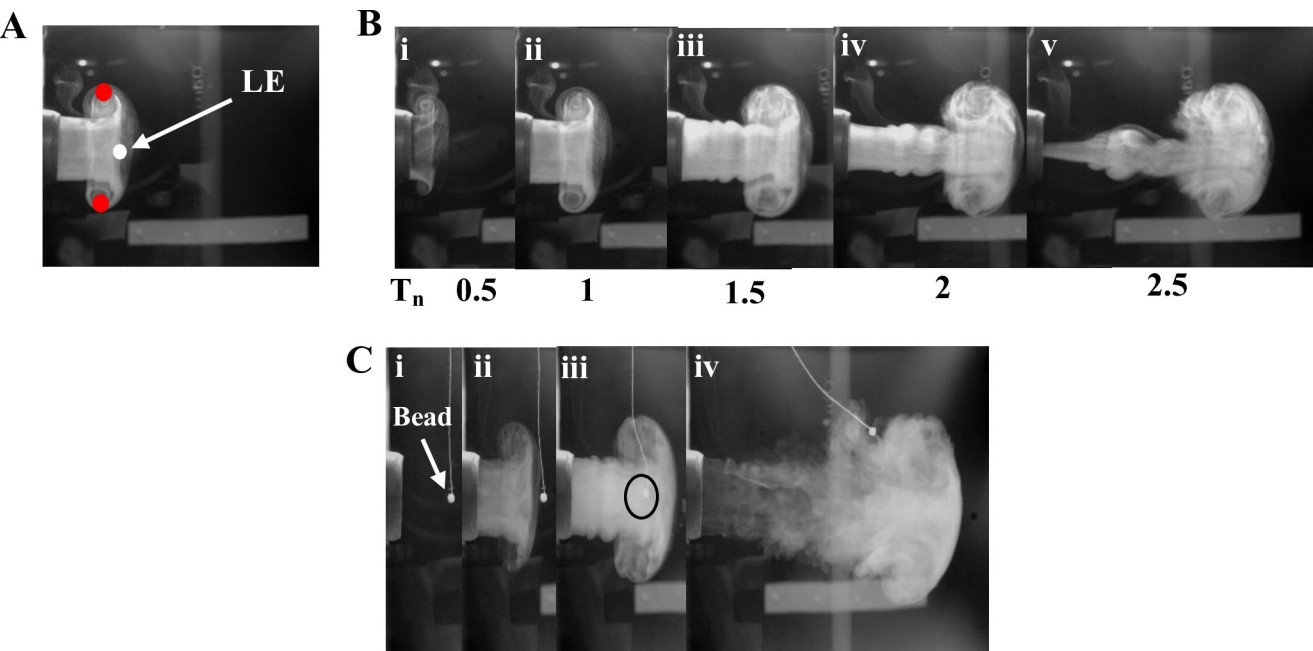

**Fig 3. Flow visualization and characterization of vortex ring.** (A) White and red circles denote leading edge (LE) and extreme end of the ring, respectively. These were tracked to calculate its axial position and diameter respectively. (B) Flow visualization at different time instances for $U_{avg}$ = 1.9 m/s. The ring propagates from left to right. $T_n$ = 0 indicates the time instance when ring just starts forming. (C) Effect of gust ($U_{avg}$ = 6.4 *m/s*) on a freely hanging Styrofoam bead. Position of bead (i) when there is no gust, (ii) just before the gust, (iii) during gust, and (iv) after gust. The bead moves with the gust (iii) until the thread is taut. Black circle in (iii) shows the position of the bead when it is at the centre of the vortex ring.

This method allowed us to make point measurements of the velocity field created by the gust. We suspended the styrofoam bead using thin sewing thread from the test chamber ceiling, such that it rested on the centreline of the nozzle exit and intercepted the vortex ring. The bead was placed at different axial locations along the centreline of the nozzle exit to measure the velocity of the bead, and hence the gust at various axial locations (Fig 3C).

Using a 12-bit CMOS camera (Phantom Miro EX4, Vision Research, Ametek, New Jersey, USA) fitted with an 18–70 mm focal length lens (Nikon, Tokyo, Japan), we recorded the flow images for both these methods at 1200 fps and 50 $\mu$s exposure time. Because of the low exposure time, we additionally illuminated the background using two 1000W halogen lamps. The camera was placed to record a lateral view of vortex ring propagation in a plane perpendicular and vertical to the nozzle exit plane. The external diameter of the nozzle served as a calibration scale for the images.

Based on exit diameter of the nozzle ($D_o$) and the ring average velocity ($U_{avg}$), we define non-dimensional time $T_n = U_{avg}t/D_0$ where $t$ is the measured time. Similarly, the axial distance $X_n$ from the nozzle exit is non-dimensionalized with the exit diameter $D_0$ and given by $X_n = X/D_0$. The dimensionless diameter of the ring is given by $D_n = D_{vb}/D_0$, where $D_{vb}$ is the instantaneous diameter of vortex bubble (i.e., diameter of the ring with entrained air; Fig 1), and dimensionless velocity of the ring is given by $U_n = U_{vb}/U_{avg}$ where $U_{vb}$ is instantaneous velocity of the vortex bubble.

## Experiments with freely-flying soldier flies

We tested this apparatus on soldier flies *Hermetia illucens* obtained from a culture housed in the National Centre for Biological Sciences campus at Bangalore. Because solider flies are

strongly attracted to light, it is possible to exploit this behavior to control the direction of their flight, which was crucial in our experiment. We released a group of 5–6 flies together into a test chamber to increase the probability that at least one of them encountered the gust. We recorded their flight motion at 4000 fps using two synchronized, high-speed cameras (Phantom VEO 640L and Phantom V611, Vision Research) for more than 80 trials, 14 of which were calibrated and digitized using MATLAB-based routines-easywand5 and DLTdv7 [30], respectively to measure their body and wing kinematics in response to the gust. The image data were down-sampled to 1000 fps, and then digitized. We tracked the head, abdomen, each wing base, and tip to get their 3D position in the global reference frame. The Centre of Mass (CoM) was assumed to lie at one-third body length from the abdomen [31] and used to represent the flight trajectory. X is the direction along the nozzle centreline, and Y and Z represent lateral and vertical axes, respectively. Here, the origin $(X,Y,Z) = (0,0,0)$ is placed at the centre of the nozzle exit, and X is positive in forward direction of the ring while Y- and Z axes conform to the conventions of the right-handed coordinate system.

The total velocity (velocity magnitude) of the fly is the resultant of three velocity components. A fourth-order low-pass Butterworth filter with cut-off frequency 200Hz was applied to the corresponding CoM data to minimize digitization error. We next calculated the velocity along each axis using a second-order central difference scheme. We non-dimensionalized velocity by dividing it by the product of the body length of flies and their wingbeat frequency. Similarly, we calculated the body roll angle $(\gamma)$ relative to the horizontal plane as the elevation angle of the vector joining the wing base and the CoM. Counter-clockwise rotations with respect to the axial direction of forward flight were treated as positive.

## Results and discussion

### Characterization of gust pertubations

The discrete gust is characterized by the spatial and temporal evolution of vortex ring and its translational velocity. We observed that the ring velocity varies linearly from 0.4 m/s to 6.4 m/s for input voltages $(V_{in})$ ranging from 2.3V to 23V ($U_{avg} = 0.28 \times V_{in}$, $R^2 = 0.99$) (Fig 4A), with Re ranging from $1 \times 10^3$ to $1.6 \times 10^4$ respectively. Such strong dependence of the translational velocity of the ring on voltage shows that by modulating the amplitude of input signal, vortex rings of different strengths can be generated, thus allowing control over flow properties. The bead velocity and the average velocity of gust measured using flow visualization matched each voltage input, thus providing cross-validation of these methods. Hence, the bead method, which is easy to establish, can be used to measure air flow. The velocity changes in a similar fashion with time and space (distance along axis) for each voltage value.

As a special case, we discuss the ring flow properties for Re = $1.6 \times 10^4$ ($U_{avg}$ = 6.4 m/s) (Fig 4B–4D). The ring propagates as a quadratic function of time until $T_n = 4$, after which it moves linearly until it reaches near the opposite wall of the test chamber (Fig 4B). It slows down, increases its diameter, and begins to deform as it approaches the opposite wall. The diameter of the ring also grows as a quadratic function of space from $1.25D_0$ to $2.25D_0$ up to $X_n = 3$ (Fig 4C), beyond which it attains space invariant final size of $2.3 \pm 0.03$ $D_0$. Because the core of the ring was not visible in all images and across all videos, we tracked the point LE to measure its axial position, and lateral extremes of vortex ring to measure its diameter (Fig 3A). The diameter measured here is, thus, the diameter of the vortex bubble and not the ring. Entrained fluid generally constitutes about 20–40% of the total volume of fluid carried by tube generated vortex ring [27]. In the present study, it was experimentally difficult to measure the entrained mass fraction. Instead, we estimated it theoretically based on the eccentricity $(e)$ of the ring.

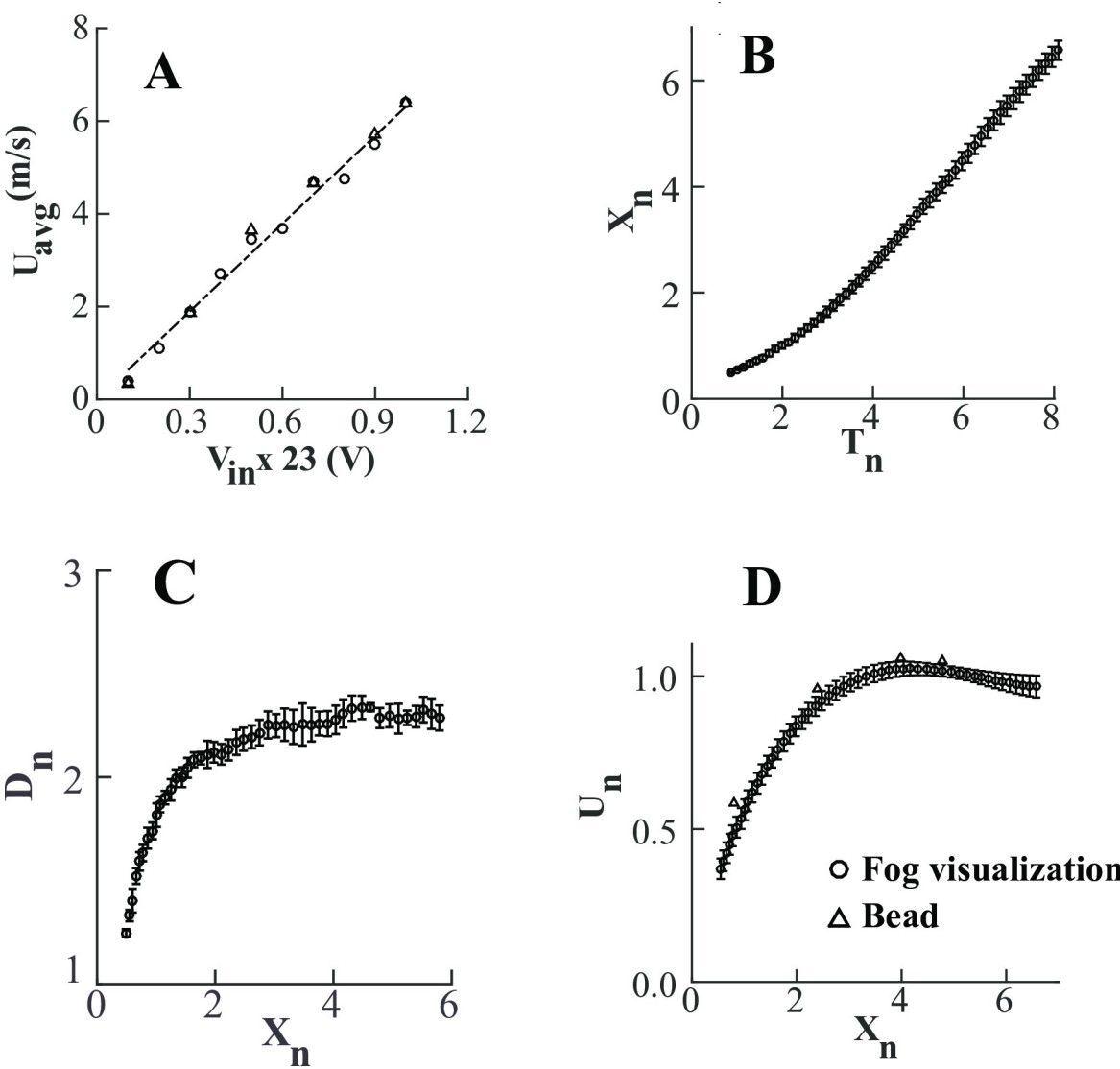

**Fig 4. Flow characteristics of vortex ring.** (A) Average propagation velocity of the ring as a function of input voltage to the speaker. Velocity of the ring obtained using fog visualization (circles) and bead method (triangles) are in good agreement for different input voltages. Dashed line is $U_{avg} = 0.2745\ V_{in}$, $R^2 = 0.99$. (B-D) Non-dimensional flow properties of vortex ring with $U_{avg} = 6.4\ m/s$ and vortex bubble diameter 8.6 cm measured using flow visualization and bead method. (B) $X_n$ is the axial distance from the nozzle exit, non-dimensionalized reative to the exit diameter $D_0$ and given by $X_n = X/D_0$. $X_n = 0$ indicates the centre of nozzle exit. (C) $D_n = D_{vb}/D_0$, is the dimensionless diameter of the ring, where $D_{vb}$ is the instantaneous diameter of vortex bubble. (D) $U_n = U_{vb}/U_{avg}$ is the dimensionless velocity of the ring, where $U_{vb}$ is instantaneous velocity of the vortex bubble. Values are mean ± SD.

For the present study, $e = 0.62$ and entrained mass fraction, $k = 0.57$. Similar values of eccentricity are observed in different studies [22, 27, 32]. $k = 0.57$ yields $D_r = 0.86 D_{vb}$.

The non-dimensional ring velocity, calculated here by applying second-order central difference scheme to axial position, becomes uniform after $X_n = 3$ from the nozzle exit (Fig 4D). The average speed of the ring for $Re = 1.6 \times 10^4$ was 6.4 m/s. The average velocity of the ring corresponds to the velocity averaged over $X_n \geq 3$ after which it is nearly constant, consistent with previous observations [33]. The maximum standard deviation in ring properties measured for three trials was less than 10% and the average standard deviation was less than 5% of their mean values, implying high repeatability of the measured values. Because the ring attains

nearly constant size and constant velocity after a particular time and distance from the nozzle and remains so for a long time and axial distance (Fig 4D), it provides a longer spatial and temporal window to study the gust response of the insects. This method can be used to study perturbation responses in both free-flying insects and hovering insects.

When using a vortex ring as a gust, ideally the gust size should match the characteristic size of the study animal. For instance, in birds, bats and insects, the ring diameter should exceed their wingspan for a head-on gust. For lateral gust perturbations in fish, it should exceed their body length. This allows their flight and swimming to be contained within the ring under controlled laboratory conditions. In our experiments, the ring diameter $D_{vb}$ (8.5 cm) was about 4 times the fly wingspan (2.2 cm). Because animal trajectories are not under the control of the experimenter, there is some spatial and temporal uncertainty about where the animal intercepts the vortex ring. For instance, in one of case studies depicted in Fig 5, the gusts due to the vortex ring hit only the right wing of one of the flies (shown in black). Furthermore, the translational velocity of the ring should match the order of forward velocity of the subject under consideration to induce an optimal response. Although this gust velocity may hold for birds, bats and fish, in case of insects that flap their wings at higher rates, the ring velocity may need to be of the order of the wing tip velocity. In the present experiments, the ring velocity was chosen to be 6.4 m/s, comparable to the wing-tip velocity of the fly (~4.85 m/s).

Another important consideration when employing vortex ring as a gust is that its formation number should be ideally less than 4 [34]. This ensures that there is no trailing jet behind the ring, and gust is contained within the ring so as not to elicit secondary response in fliers. In all the experiments reported here, the formation number was ≈3.2 (see calculation in S1 Appendix).

## Vortex ring based gust perturbation in flying insects

To demonstrate the effectiveness of the vortex ring as a precise gust-imparting device, we subjected freely-flying soldier flies *Hermetia illucens* with head-on gusts, and measured the influence of these gusts on their flight trajectory, velocity and body angles. As mentioned above, we selected vortex ring velocity of 6.4 m/s ($Re = 1.6 \times 10^4$) to perturb the flies. We have chosen results from four experiments to illustrate the types of responses that were obtained.

In these experiments, the flies were perturbed by the gust at mean axial location of $X_0 = 3.7$ ±0.38 $D_0$ from the exit of the nozzle, and were always contained inside the gust at encounter (Fig 5A, and S2 Video). If the flies flew to the left side of the gust, they continued to fly on the same side even after being hit by the gust. Similarly, they flew downward after the encounter with gust, indicating a possible loss in lift forces. The flies did not recover their initial vertical and lateral position after gust in any trial. Even in trials for which the gust was not visible, the time instant of the gust encounter could be precisely determined by the large displacement of the antennae (S2 Video).

Before being hit by the ring, the forward speed of the flies was near-constant in each case. However, the gust decreased their forward speed by as much as ~70% maximum and ~30% on average (Fig 5B). The deceleration of flies due to gust in the present study is consistent with observation in bumble bees [15, 18], indicating gust as a limiting parameter on the forward speed of insects [4]. On the other hand, recently, researchers have found an increase in airspeed of pigeon when flying in turbulence [35], and golden eagles exploiting gust to accelerate [36], suggesting that birds might enhance their airspeed by harnessing energy from the surroundings.

The flies had near-zero body roll angle (<15˚) before intercepting the vortex ring, but changed by as much as (~160˚) due to gust (Fig 5C, and S2 Video). Insect such as honeybees

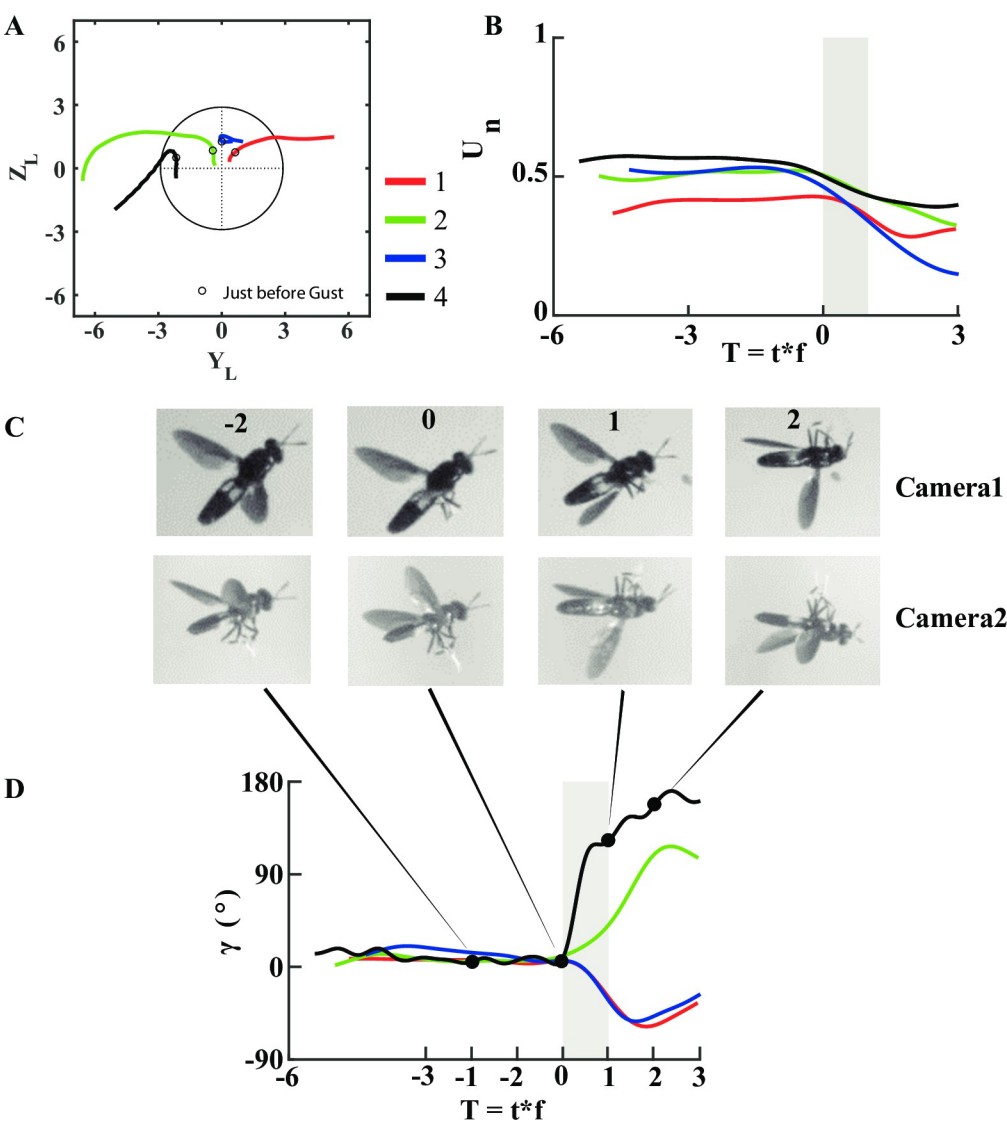

**Fig 5. Changes in body kinematics of soldier flies (*Hermetia illucens*) due to the gust induced by the vortex ring in 4 different trials.** (A) Trajectory in Y-Z plane normalized by the average body length ($L_{avg}$) of flies. The mean axial distance where the gust hits the insects is $X_0 = 3.7 \pm 0.38 D_0$. Black circle indicates the front view of the vortex ring, and the intersection of vertical and horizontal dashed lines is the centre of the vortex ring. Coloured lines are the trajectories of flies for each trial represented by 1–4, and open circles on each curve denote the position of the flies just before they were hit by the ring. (B) Normalized speed versus non-dimensional time (in wing beats). Forward speed is normalized with the average speed of the flies before being hit by the ring, and time is non-dimensionalized by multiplying with the wing beat frequency. T = 0 indicates the time instance just before flies were just hit by the ring. Time period of gust is indicated in vertical grey strip. (C) Oblique top and the corresponding side views of flight sequences for trial 4 at different time instances showing distinct change in body roll angle. Number on the top row indicates the time instance of fly with respect to gust in terms of wing beats. (D) Change in body roll angle plotted against the wingbeat. Filled circles on black curve (trial 4) denote the time instances of the fly in (C).

[18], bumblebees [8, 15] and orchid bees [4] have also been observed to respond highly to gusts along roll axis. High roll rates of body and tail are also observed in ruby-throated humming birds when flying in turbulent wind conditions rather than undisturbed air [14]. Some insects extend their legs tin response to the perturbation due to gusts [4, 18], whereas ruby-throated hummingbirds compensate for turbulence by modulating their wing kinematics

including stroke angle, stroke and asymmetry [14]. In the present study, soldier flies extended their legs as well as changed their wing stroke angles during recovery from the gusts. The observed changes in the flight trajectory, speed and body orientation show that the vortex ring can be used as a precise gust to perturb the trajectories of flies.

Although this method provides a well-characterized and precise gust, the actual response of the insects depends on their radial position at the time they are hit by the vortex ring. These conditions can be well-controlled in robotic flappers or drones, but there is no easy way to ensure that freely-flying insects will have a repeatable position and wing configuration at the time of impact. Because the axial velocity within the ring is non-uniform, it is difficult to elicit repeatability in their response of the insects. As the gust dynamics are predictable, a post-hoc reconstruction of the impact of gust is possible provided the fly does not intercept gust towards the rim of the vortex ring. Thus, this method greatly enhances our ability to determine the magnitude of the gust encounter and interpret the response of the fly relative to this gust. Overall, this method can lend itself to studies on the biomechanics and control and stability of diverse animals and robotic fliers.

## Conclusion

We present a method of gust generation via a discrete vortex ring. The ring was generated by an impulsive motion of a diaphragm of an electronic speaker. The flow physics of this perturbation method is well-understood and hence, the flow properties are highly controllable. Besides, this method allows high repeatability and reproducibility, can be implemented at a low-cost, and has a simple mechanical design. As an example, we tested its application to study the impact of gust on insect flight, but as such this method can be used in a variety of situations and habitats. The application of the vortex ring as a gust is not only limited to insects but could potentially be extended to study birds, bats, and micro-aerial vehicles (MAVs) in air, and fish and underwater autonomous vehicles in water. We have provided relevant theoretical relations that will be useful for design of a gust generator for a specific application.

## Supporting information

**S1 Video. Gust generation and characterization.**
(MP4)

**S2 Video. Insect and gust interaction.**
(MP4)

**S1 Appendix.**
(DOCX)

## Acknowledgments

We thank Dr Toshiyuki Nakata, Chiba University for his helpful feedback on this manuscript.

## Author Contributions

**Conceptualization:** Dipendra Gupta, Sanjay P. Sane, Jaywant H. Arakeri.

**Data curation:** Dipendra Gupta.

**Formal analysis:** Dipendra Gupta, Sanjay P. Sane, Jaywant H. Arakeri.

**Funding acquisition:** Sanjay P. Sane, Jaywant H. Arakeri.

**Investigation:** Dipendra Gupta, Jaywant H. Arakeri.

**Methodology:** Dipendra Gupta, Sanjay P. Sane, Jaywant H. Arakeri.

**Project administration:** Sanjay P. Sane, Jaywant H. Arakeri.

**Resources:** Sanjay P. Sane, Jaywant H. Arakeri.

**Supervision:** Sanjay P. Sane, Jaywant H. Arakeri.

**Validation:** Dipendra Gupta.

**Visualization:** Dipendra Gupta, Sanjay P. Sane.

**Writing – original draft:** Dipendra Gupta, Sanjay P. Sane, Jaywant H. Arakeri.

**Writing – review & editing:** Dipendra Gupta, Sanjay P. Sane, Jaywant H. Arakeri.

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
