## [Decision Letter · Decision Letter 0]

21 Feb 2024

PONE-D-24-02005Generating controlled gust perturbations using vortex ringsPLOS ONE

Dear Dr. Sane,

Thank you for submitting your manuscript to PLOS ONE. After careful consideration, we feel that it has merit but does not fully meet PLOS ONE’s publication criteria as it currently stands. Therefore, we invite you to submit a revised version of the manuscript that addresses the points raised during the review process.

As you can see both reviewers state that this is a technically sound study. However, there are several questions the reviewers have bought up which need to be addressed before the paper can be accepted for publication.

We look forward to receiving your revised manuscript.

Kind regards,

Iman Borazjani, Ph.D.

Academic Editor

PLOS ONE

“Funding for this study was provided by grants from the Air Force Office of Scientific Research (AFOSR) # FA2386-11-1-4057 and # FA9550-16-1-0155, and National Centre for Biological Sciences(Tata Institute of Fundamental Research) to SPS. We also acknowledge the support of the Ministry of Earth Sciences, Government of India, under project no. MESO-0034 and the Department of Atomic Energy, Government of India, under project no. 12-R&D-TFR-5.04-0800.”

“Funding for this study was provided by grants from the Air Force Office of Scientific Research

 (AFOSR) # FA2386-11-1-4057 and # FA9550-16-1-0155, and National Centre for Biological

Sciences(Tata Institute of Fundamental Research) to SPS. We also acknowledge the support of

the Ministry of Earth Sciences, Government of India, under project no. MESO-0034 and the

Department of Atomic Energy, Government of India, under project no. 12-R&D-TFR-5.04-

0800.”

“Funding for this study was provided by grants from the Air Force Office of Scientific Research (AFOSR) # FA2386-11-1-4057 and # FA9550-16-1-0155, and National Centre for Biological Sciences(Tata Institute of Fundamental Research) to SPS. We also acknowledge the support of the Ministry of Earth Sciences, Government of India, under project no. MESO-0034 and the Department of Atomic Energy, Government of India, under project no. 12-R&D-TFR-5.04-0800.”

5. In this instance it seems there may be acceptable restrictions in place that prevent the public sharing of your minimal data. However, in line with our goal of ensuring long-term data availability to all interested researchers, PLOS’ Data Policy states that authors cannot be the sole named individuals responsible for ensuring data access (http://journals.plos.org/plosone/s/data-availability#loc-acceptable-data-sharing-methods).

Reviewers' comments:

Reviewer's Responses to Questions

**Comments to the Author**

1. Is the manuscript technically sound, and do the data support the conclusions?

Reviewer #1: Yes

Reviewer #2: Yes

2. Has the statistical analysis been performed appropriately and rigorously? 

Reviewer #1: I Don't Know

Reviewer #2: N/A

3. Have the authors made all data underlying the findings in their manuscript fully available?

Reviewer #1: Yes

Reviewer #2: Yes

4. Is the manuscript presented in an intelligible fashion and written in standard English?

Reviewer #1: Yes

Reviewer #2: Yes

5. Review Comments to the Author

Reviewer #1: The authors present a study of generating gusts using vortex rings and have applied it to study response of insects to such gusts. In general the paper is written well, however, there are a few points to be addressed

- The vortex characterization is missing any discussion on the formation number of the vortex rings generated. This number is crucial to understand what kind of structure is formed. “A universal time scale for vortex ring formation” (Gharib 1998) is a very important reference regarding vortex ring generation. Given the rings are generated with constant time pulsing and different Reynolds numbers are produced, the vortex rings will have different formation numbers. This difference means the structure of the gust will fundamentally change for each Reynolds number, defeating the whole purpose of the desired well-characterized and repeatable gust.

- Furthermore, If the formation number is larger than 4, then not all of the fluid driven by the piston, or in this case speaker, will roll up into the vortex ring, or bubble as it is called in this paper, so any assumptions like equation 7 and the mass fraction relationship in the appendix are potentially flawed. Since there will be a trailing jet which can make the vortex unstable, it might not present a repeatable case of gust generation using vortex rings at such high formation numbers. Have the authors tried lower formation numbers to ensure better repeatability of the vortex ring formation and hence the associated gust?

- Could the authors comment on the efficacy of using a bead to estimate the velocity of the gust, perhaps there are better ways of achieving this end outcome?

- The authors could take this opportunity to discuss some more about the flight dynamics of the insect under such perturbations. Perhaps estimate some aerodynamics on the flapping wing and comment on the aerodynamic loading and associated responses of the insect. Also comparing these responses to other natural flyers such as birds (across several flapping time scales - eagles/owls to hummingbirds) during gust encounters.

Additional specific comments are below

- Lines 67-70: Suggest rewording with Vortex ring as the subject as that is what is important, not the impulsive flow. The sentence reads awkwardly with that as the subject. “Vortex rings are generated from…”

- Lines 94-95: Saying the piston movement generates the layer of vorticity is awkward. The piston drives a slug of fluid, and it is the motion of this slug combining with the no-slip condition to form a boundary layer. This is important especially in some designs when the piston is far upstream of the nozzle.

- Line 96: As this is a general description of a vortex generation, I would say fluid instead of air.

- Lines 134-135: Many vortex ring studies are not concerned with the stopping vortex. If it is important to prevent from forming in this application, then provide explanation as to why it is important.

- Line 144: missing “a” before “long nozzle”

- Lines 145-146: missing “a” or “the” before piston vortex

- Line 146: I believe the antecedent of “that” is the plural “disturbances,” so replace with “to those generated using an orifice.”

- Line 149: Why have the authors chosen to put “e.g.” in their citation just this one time? Is it more applicable than the other citations?

- Line 170: What is meant by “at different points in time?” Either delete or provide more detail.

- Line 171: The faster vortex ring is referenced to Figure 3B, and the slower vortex ring is referenced to Figure 3C, but the caption is backwards. Please make the caption and reference agree.

- Lines 131-177

- Lines 410-411 The wording defining =0 as the “time instance when no ring is formed” is awkward. Defining that time as the last frame where there is no flow or the instant the vortex generation process begins is clearer.

Reviewer #2: This work introduces a system to generate controllable gusts relying on vortex rings for studying animal locomotion. The authors provided detailed design and characterization of the gust generator, and its application for a case study on free-flying soldier flies. In general, I find the paper is very interesting, very well written, and easy to follow. It also provide a useful and easy-to-build method for studying insect and insect-inspired flight in gusty conditions. I have several minor comments for the authors:

1) It is still unclear what is the limitations of existing methods of gust generation in the Introduction (lines 59-64)

2) I recommend to add a sub-title for the result on gust characterization, for example “Generation of gust perturbations”

3) Line 262: “When using a vortex ring as a gust, it should fully encompass the subject.” Why is this a condition? What happen if it cannot cover the subject? For example, the flight trial 4 of the fly in Movie 2 may represent this case as the gust hit the right wing only (or impact stronger on the right wing), causing rolling response of the fly.

6. PLOS authors have the option to publish the peer review history of their article (what does this mean?). If published, this will include your full peer review and any attached files.

Reviewer #1: No

Reviewer #2: No

---

## [Author Response · Author response to Decision Letter 0]

15 Apr 2024

General comments:

We thank both reviewers for their engagement with the manuscript and for providing detailed and helpful comments, which we have tried to address below. The suggested changes add much value and rigor to the manuscript, and also help streamline it a little better. In the responses of reviewers below, the referee’s comments are in bold, and our responses are in regular font. The changes made in the manuscript have been highlighted in yellow.

Response to Reviewer #1:

-The vortex characterization is missing any discussion on the formation number of the vortex rings generated. This number is crucial to understand what kind of structure is formed. “A universal time scale for vortex ring formation” (Gharib 1998) is a very important reference regarding vortex ring generation. Given the rings are generated with constant time pulsing and different Reynolds numbers are produced, the vortex rings will have different formation numbers. This difference means the structure of the gust will fundamentally change for each Reynolds number, defeating the whole purpose of the desired well-characterized and repeatable gust.

We agree that the paper would benefit by the inclusion of some discussion of the formation number, and have therefore added text to clarify the role of formation number in our method. We have also included the citation of Gharib (1998) in this context. Because we used a speaker to generate vortex rings, the calculation of a formation number was somewhat trickier, but we were able to do so using measured characteristics of the vortex rings. We elaborate on this point below:

For small formation numbers (<4), the vortex ring is formed with all the vorticity and mass being entrained from the slug of fluid exiting the nozzle, and the ring achieves its maximum size. As formation number increases, the size and strength of the ring do not increase. The additional fluid mass and vorticity is contained in the trailing jet, giving rise to “pinched-off” state, thus altering the structure of the flow. 

In the present study, the experiments with soldier flies were conducted at the same Reynolds number (Re=(U_avg D_0)/ν=1.6 × 〖10〗^4), and hence also same formation number. The speed or diameter of the vortex rings was left unaltered. Because the experiments were carried out in a closed room at ambient temperature of 〖20〗^∘ C, i.e. at constant kinematic viscosity, Re based on the speed and size of the ring and the kinematic viscosity of air remained constant in all the experiments with soldier flies. 

During characterization of the rings, we generated vortex rings with various speeds, and hence, different Re (see fig 4A). Fig. 3B is one such case showing the formation of the ring at Re=4.7×〖10〗^3 but did not use the low Re vortex ring to probe the response of soldier flies.

All the experiments with soldier flies were conducted at formation number ≈ 3.2 (as estimated below) which was the maximum formation number for the vortex rings in our experiments. This is consistent with the fact that, qualitatively, we did not observe a trailing edge pinch-off in the flow visualization of the ring (See SI video1). Moreover, a secondary response would be expected from the flies if they were hit first by the ring and then again by a trailing jet formed at formation numbers > 4, but we never observed such a response. These observations strongly suggest that our formation number does not exceed 4. 

We estimated the formation numbers in our setup in the following way:

Estimation of formation number:

In the experiments reported here, Dr = 0.86Dvb=7.3 cm, e =0.62, and D0 =3.7 cm. Inserting these values into equation 4, 

L=(2D_r^3 e)/(3D_o^2 )=11.75 cm

And the formation number, L/D_(0 )=3.17 

This estimate is based on the diameters of the ring and the nozzle exit, which are measured quantities in this study, and additionally assumes volume conservation (i.e., all the fluid ejected from the nozzle gets rolled into the vortex ring) based on flow visualization and SI Video1. 

- Furthermore, If the formation number is larger than 4, then not all of the fluid driven by the piston, or in this case speaker, will roll up into the vortex ring, or bubble as it is called in this paper, so any assumptions like equation 7 and the mass fraction relationship in the appendix are potentially flawed. Since there will be a trailing jet which can make the vortex unstable, it might not present a repeatable case of gust generation using vortex rings at such high formation numbers. Have the authors tried lower formation numbers to ensure better repeatability of the vortex ring formation and hence the associated gust?

We agree that equation 7 will not hold if the formation number exceeds 4, and have added this cautionary statement in line 381 -382. In our case, the vortex rings were generated at lower formation numbers (see Fig. 4A), and each case was repeatable. Fig. 4 B-D shows mean values ± standard deviation (which is less than 5% the mean values) for the vortex ring used in the present study, and serves a good check for repeatability of the vortex ring and the associated gust. 

- Could the authors comment on the efficacy of using a bead to estimate the velocity of the gust, perhaps there are better ways of achieving this end outcome?

This study uses two methods to estimate the gust velocity. First, we use a Styrofoam bead because its density is of the same order as that of air, i.e., it is close to neutrally buoyant. This means that it instantaneously responds to the flow, and can be safely used as a tracer to measure the average gust velocity. Second, we used the fog visualization method to corroborate the bead measurement. There is an excellent agreement between the velocity estimate obtained using the bead and that calculated using fog visualization method for different speed of gusts (Fig 4A, D). An added advantage of using bead is that it is economical and readily available in the local market, which allowed us to get a rough estimate of the gust propagation speed quickly, especially we hope that this method will be readily accessible.

- The authors could take this opportunity to discuss some more about the flight dynamics of the insect under such perturbations. Perhaps estimate some aerodynamics on the flapping wing and comment on the aerodynamic loading and associated responses of the insect. Also comparing these responses to other natural flyers such as birds (across several flapping time scales - eagles/owls to hummingbirds) during gust encounters.

As suggested by the reviewer, we have added a brief discussion in the main text (line 308 to 321) on comparison of the observed responses in soldier flies to other natural flyers. 

Additional specific comments are below

- Lines 67-70: Suggest rewording with Vortex ring as the subject as that is what is important, not the impulsive flow. The sentence reads awkwardly with that as the subject. “Vortex rings are generated from…”

Agreed. We have made the suggested changes. 

- Lines 94-95: Saying the piston movement generates the layer of vorticity is awkward. The piston drives a slug of fluid, and it is the motion of this slug combining with the no-slip condition to form a boundary layer. This is important especially in some designs when the piston is far upstream of the nozzle.

Agreed. We have made the suggested changes. 

- Line 96: As this is a general description of a vortex generation, I would say fluid instead of air.

Agreed. We have made the suggested changes.

- Lines 134-135: Many vortex ring studies are not concerned with the stopping vortex. If it is important to prevent from forming in this application, then provide explanation as to why it is important.

We have made the suggested changes.

- Line 144: missing “a” before “long nozzle”

We have made the suggested changes.

- Lines 145-146: missing “a” or “the” before piston vortex

We have made the suggested changes.

- Line 146: I believe the antecedent of “that” is the plural “disturbances,” so replace with “to those generated using an orifice.”

We have replaced “that” with “those”, as suggested. 

- Line 149: Why have the authors chosen to put “e.g.” in their citation just this one time? Is it more applicable than the other citations?

We have removed ‘e.g.’ in the citation, and added more details. In the previous version, we had added e.g., because this point has been made by several other authors as well.

- Line 170: What is meant by “at different points in time?” Either delete or provide more detail.

We thank the reviewer for pointing out this error, and have deleted this.

- Line 171: The faster vortex ring is referenced to Figure 3B, and the slower vortex ring is referenced to Figure 3C, but the caption is backwards. Please make the caption and reference agree.

Thanks much for pointing out this error. We have corrected it.

- Lines 410-411 The wording defining =0 as the “time instance when no ring is formed” is awkward. Defining that time as the last frame where there is no flow or the instant the vortex generation process begins is clearer.

We have made the changes, as suggested. 

Response to Reviewer #2:

1) It is still unclear what is the limitations of existing methods of gust generation in the Introduction (lines 59-64)

One advantage of using vortex ring as a gust is that it is possible to generate a high-speed perturbation while keeping the ring laminar. Laminarity is important to eliminate any ambiguity in the flow characteristics, and consequently, to better understand the response of insects. On the contrary, most of the existing methods are turbulent gusts. For example, jets and grid-generated turbulence are turbulent, and therefore, the response of the insects could be a function of both mean flow and turbulent fluctuations, thereby making it difficult to decouple their effects on the insects’ response. Karman-vortices, on the other hand, are generated by cylinders placed in the upstream, which lead to wake formation, and hence, lack of neat and laminar flow in the region where insect experiences gusts. Vortex ring is, however, free from these effects. We have added these points in the main text (line 73-79).

2) I recommend to add a sub-title for the result on gust characterization, for example “Generation of gust perturbations”

We have added the suggestion sub-title.

3) Line 262: “When using a vortex ring as a gust, it should fully encompass the subject.” Why is this a condition? What happen if it cannot cover the subject? For example, the flight trial 4 of the fly in Movie 2 may represent this case as the gust hit the right wing only (or impact stronger on the right wing), causing rolling response of the fly.

We thank the reviewer for this question, and have, accordingly, made the changes (line 277-280).

---

## [Decision Letter · Decision Letter 1]

24 May 2024

Generating controlled gust perturbations using vortex rings

PONE-D-24-02005R1

Dear Dr. Sane,

We’re pleased to inform you that your manuscript has been judged scientifically suitable for publication and will be formally accepted for publication once it meets all outstanding technical requirements.

Kind regards,

Iman Borazjani, Ph.D.

Academic Editor

PLOS ONE

Additional Editor Comments (optional): Reviewer 1 has a minor comment, which can be addressed when submitting all the finals or during the proofing. 

Reviewers' comments:

Reviewer's Responses to Questions

**Comments to the Author**

1. If the authors have adequately addressed your comments raised in a previous round of review and you feel that this manuscript is now acceptable for publication, you may indicate that here to bypass the “Comments to the Author” section, enter your conflict of interest statement in the “Confidential to Editor” section, and submit your "Accept" recommendation.

Reviewer #1: (No Response)

Reviewer #2: All comments have been addressed

2. Is the manuscript technically sound, and do the data support the conclusions?

Reviewer #1: Yes

Reviewer #2: Yes

3. Has the statistical analysis been performed appropriately and rigorously? 

Reviewer #1: I Don't Know

Reviewer #2: (No Response)

4. Have the authors made all data underlying the findings in their manuscript fully available?

Reviewer #1: No

Reviewer #2: Yes

5. Is the manuscript presented in an intelligible fashion and written in standard English?

Reviewer #1: Yes

Reviewer #2: Yes

6. Review Comments to the Author

Reviewer #1: - Appendix line 384 - "... if the formation number (L/D0) is approximately 4 (34)".. should this be "... (L/D0) is limited to approximately 4 (34)" ?

Reviewer #2: The authors have reflected all my concerns. The reviewer has no further comments and recommends acceptance.

7. PLOS authors have the option to publish the peer review history of their article (what does this mean?). If published, this will include your full peer review and any attached files.

Reviewer #1: No

Reviewer #2: No

---

## [Editor Report · Acceptance letter]

31 May 2024

PONE-D-24-02005R1 

PLOS ONE

Dear Dr. Sane, 

I'm pleased to inform you that your manuscript has been deemed suitable for publication in PLOS ONE. Congratulations! Your manuscript is now being handed over to our production team.

Kind regards, 

on behalf of

Dr. Iman Borazjani 

Academic Editor

PLOS ONE